# Mobile Phone Addiction and Sleep Quality: The Mediating Role of Anxiety and the Moderating Role of Emotion Regulation

**DOI:** 10.3390/bs13030250

**Published:** 2023-03-12

**Authors:** Li Gong, Qiang Liu

**Affiliations:** 1Research Center of Brain and Cognitive Neuroscience, Liaoning Normal University, Dalian 116029, China; 2Institute of Brain and Psychological Sciences, Sichuan Normal University, Chengdu 610000, China

**Keywords:** mobile phone addiction, sleep quality, anxiety, emotion regulation, cognitive reappraisal, expressive suppression

## Abstract

Smartphones have become a fundamental tool in the daily life of mankind, but its excessive use seriously impairs people’s quality of sleep. A specific state of emotion has been shown to play a crucial role in the relationship between mobile phone addiction (MPA) and the sleep quality of college students. However, studies have rarely considered top-down emotion regulation. This study is the first to examine the effects of MPA on the sleep quality of Chinese college students from a top-down emotion regulation perspective. The survey sample comprised 1559 university students (40.73% male; M (SD) age = 19.11 (1.22) years) who completed questionnaires on MPA, sleep quality, anxiety and emotion regulation. The results revealed that (1) Anxiety mediated the relationship between MPA and sleep quality; (2) Cognitive reappraisal (CR) negatively moderated the relationship between MPA and anxiety; and (3) Expressive suppression (ES) positively moderated the relationship between MPA and anxiety. These findings reveal the mechanism of sleep problems in Chinese college students. We provide research ideas and method guidance for the follow-up intervention and treatment of college students’ sleep problems.

## 1. Introduction

With the rapid development of communication technology, the smartphone, as an information carrier, has penetrated into people’s daily life, and mobile phone addiction (MPA) is its largest negative product [1]. MPA refers to addictive behavior in which individuals use their phones excessively and compulsively, thereby negatively affecting their psychological and social functions [2,3]. Research has found that MPA not only adversely affects people’s social and psychological functioning, such as in their interpersonal relationships [4], emotion and cognition [5], but that it can also affect people’s sleep quality and other physiological functions [6,7].

Sleep quality is a comprehensive evaluation of sleep efficiency, sleep disturbance and daytime function [8]. Previous studies have shown that the degree of MPA can directly or indirectly predict sleep quality. The indirect path is mainly through psychological and physiological aspects. From the psychological aspect, the excessive use of cell phones can consume a lot of energy and seriously affect students’ learning and concentration during the day [9]. It then induces anxiety and reduces the quality of sleep at night [10]. Furthermore, Qqla et al. [6] found that MPA can reduce sleep quality by affecting people’s cognitive levels. From the physiological aspect, screen light from electronic devices can affect an individual’s melatonin production, which interferes with people’s sleep rhythms and affects their quality of sleep [11].

Sleep problems have serious effects on college students’ learning and life, physical or mental health [12,13]. Exploring the impact of MPA on sleep quality is essential for the growth and development of students. According to the flow theory [14], paying excessive and involuntary attention to cell phone information decreases the self-awareness and time perception of cell phone addicts. During this period, they have a pleasant psychological experience [15], but then they have more negative emotions, such as anxiety [16] and depression [17]. Specifically, the overuse of mobile phones takes up a lot of people’s time and energy, exposing them to more interpersonal problems, and generates stronger negative emotions [4,18]. These show that MPA is an important trigger of anxiety in college students. In addition, the emotion regulation theory suggests that an individual’s internal emotional experience is closely related to his actual emotional perceptions and external behaviors [19]. Therefore, persistent and excessive anxiety may be an important factor leading to worse sleep quality [20,21,22]. For example, a study by Jin et al. [20] found that anxiety significantly predicted the sleep quality of individuals. Furthermore, at the genetic level, Gregory et al. [23] found through empirical research that the genes of anxiety and sleep disorders overlap, which confirmed the internal relationship between anxiety and sleep quality. Moreover, Wang et al. [22] discovered that anxiety plays crucial role in the relationship between MPA and sleep quality. Their study showed that when college students spend a lot of time on their mobile phone, this can bring physical discomfort, which, in turn, impairs their nighttime sleep. In summary, MPA may increase college students’ anxiety levels and thus impair their sleep quality.

However, it is insufficient to only consider the effect of a single emotion on the association between MPA and sleep quality. Overall, emotions play a complicated role. Emotions can stimulate individuals to respond accordingly in their behavior [24], adjust their decision-making processes [25], deepen memories of important events, and facilitate interpersonal communication [26,27]. At different stages of emotion onset, people need to adopt different emotion regulation strategies to adjust appropriately [28]. Research has shown that individuals using different emotion regulation strategies produce extremely different emotion perceptions and emotional responses [29]. Thus, in order to more comprehensively reveal the mechanism of the impact of emotion on the relationship between MPA and sleep quality, it is necessary to explore the role of top-down emotion regulation.

The process by which emotions are produced, experienced and expressed is called emotion regulation [28], and this process involves a variety of regulatory strategies. Two of these strategies, cognitive reappraisal (CR) and expressive repression (ES), are the most frequently used [30]. CR diminishes the impact of negative events by altering the comprehension of the emotional event early in its generation, thereby reducing negative emotional experiences. It is an antecedent-focused strategy [19]. Numerous studies have confirmed that CR is a positive emotion regulation strategy that improves an individual’s emotional experience [31,32,33]. Resource protection theory states that when individuals are in a state of emotional implosion, instinctive self-protection mechanisms are activated to mitigate negative effects on the self and reduce self-loss [34]. Specifically, MPA triggers anxiety in individuals [35]. However, individuals who are adept at using CR strategies can effectively reduce their negative emotional experiences and resource consumption by reappraising emotional situations early in their emotional onset [36]. It is thus evident that the use of CR strategies can effectively regulate negative emotions. Therefore, we suggest that CR can reduce the anxiety caused by MPA and thus improve the sleep quality of individuals.

However, ES is a response-focused strategy that changes one’s reaction to negative emotions by suppressing the emotions that are going to be or are being expressed [19]. This strategy is different from CR. It directly affects the physiological, experiential or behavioral reaction of emotion in the later stage of emotion occurrence [37]. Brain imaging studies have found that when people use ES strategies for emotion regulation, functional connections between the prefrontal, temporal, and hippocampal parahippocampal gyrus are enhanced, which contributes to their negative emotions [32]. The construct theory of positive emotions indicates that negative emotions can hinder the construction of mental resources in individuals [38]. This suggests that if individuals employ an ES strategy, repression controls the release of negative emotions, which seriously affects their physical and mental health [39]. Furthermore, as a negative emotion regulation strategy, more sleep problems may be seen in people who regularly use ES strategies [40,41]. A main reason for this is that the people who use ES strategies inhibit the expression of undesirable emotions, but do not make the negative emotional experience disappear completely, they just forcefully suppress it [19]. From this, it is clear that using ES strategies to regulate negative emotions may be counterproductive. Therefore, we suggest that ES has the opposite moderating effect of CR. ES may enhance the anxiety caused by MPA, which, in turn, may further impair the quality of sleep of individuals.

In summary, this study aims to investigate the impact of MPA on the sleep quality of college students, as well as to reveal the role of anxiety and top-down emotion regulation in this process, with the goal of providing feasible recommendations to improve students’ mental health. According to the above, this research proposes to construct anxiety as a mediating variable and emotion regulation as a moderating variable, in order to further explore the internal mechanisms by which MPA affects sleep quality in college students. Therefore, we proposed the following hypotheses to examine (H1) whether anxiety mediated the associations between MPA and sleep quality, (H2a) whether CR moderated the relationships between MPA and anxiety, and (H2b) whether ES moderated the relationships between MPA and anxiety; see Figure 1.

## 2. Method

### 2.1. Participants

A cross-sectional design and convenience sampling method was employed to recruit college students from 7 universities in China to conduct an online questionnaire survey. We collected 1769 questionnaires and obtained 1559 valid questionnaires after excluding those with obvious regular, random or wrong answers. The effective rate was 88.13%. Of these, 635 (40.73%) were male, 924 (59.27%) were female, 664 (42.58%) were freshman, 521 (33.42%) were sophomore, 226 (14.50%) were junior, and 148 (9.50%) were in their fourth year or more. There were 806 singleton children (51.70%) and 753 non-singleton children (48.30%). They were aged 17 to 24 (mean age = 19.11 ± 1.22 years). Considering that some of our participants were potentially under the age of 18, we added a question to the informed consent form, “If you are under the age of 18, did you get your parents’ consent to participate in this study?”.

### 2.2. Measures

Mobile Phone Addiction. The MPA scale used in this study was designed by Su et al. [42]. This scale consists of 22 items, which are divided into 6 factors: withdrawal behavior (e.g., “I’m fidgeting without my mobile phone”), social pacification (e.g., “I prefer to chat on mobile rather than face to face”), salience behavior (e.g., “My friends and family complain about my use of mobile phones”), negative influence (e.g., “Playing with smart phones affects my academic performance”), App use, and App update. The scale adopted a 5-point scoring method, which ranges from strongly disagree to strongly agree. The degree of addiction was determined by the total score, with higher scores indicating a stronger degree of addiction. The Cronbach’s alpha coefficient of the original scale was 0.88. In this study, Cronbach’s alpha coefficient of the scale was 0.94.

Anxiety. In the present study, we used the Self-rating anxiety scale (SAS) designed by Zung [43]. This scale contains a total of 20 items and uses a 4-point scale from 1 (little or no time) to 4 (most or all of the time), with 5 questions being reverse scored (e.g., “I think everything is fine”). The scale collects information about the individual’s situation in the week prior and the scores on the scale reflect the individual’s level of anxiety; the scores were proportional to the level of anxiety. The Cronbach’s alpha coefficient of the original scale was 0.93. The Cronbach’s alpha of this scale in our study was 0.85.

Emotion Regulation. This survey used the Chinese version of the Emotion Regulation Questionnaire (ERQ) adapted by Wang et al. [44]. The original authors of the scale are Gross and John [20]. There were 10 items in the questionnaire, which were divided into two factors: CR (e.g., “I will control my emotions by changing the way I think about the situation”) and ES (e.g., “I will not show my feelings”). The former includes 6 items and the latter has 4 items. The higher the score on each subscale, the more frequently the strategy is used. The Cronbach’s alpha coefficient of the original scale CR was 0.85, and ES was 0.77 [44]. The Cronbach’s alpha of the total scale in the present research was 0.78 (CR was 0.88, and ES was 0.71).

Sleep Quality. The Chinese version of the Pittsburgh Sleep Quality Index (PSQI) [45] scale consists of 18 items [8]. This scale can be divided into 7 factors, such as sleep duration, daytime functioning, sleep efficiency and subjective sleep quality, which are used to understand the individual’s sleep status in the most recent month. The scale used 4-point scoring (0~3), and the accumulated scores of each factor were the total scores. A higher score meant poorer sleep quality. The Cronbach’s alpha coefficient of the original scale was 0.85 [8]. In this study, the Cronbach’s alpha coefficient was 0.84.

### 2.3. Statistical Analysis

We used IBM SPSS 24.0 (The software was developed by SPSS, Chicago, IL, USA) to carry out descriptive statistics and a Pearson’s correlation analysis on the data. The variables in this study were all continuous variables. The present study was based on the principle of linear regression, and used PROCESS macro procedure to examine the mediating effect of anxiety between MPA and sleep quality, and the moderating effect of CR and ES on the mediating role of anxiety [46]. Because mediation effect estimates generally do not follow a normal distribution, a bootstrap sample of 5000 was constructed and confidence intervals for the mediating effects were estimated using a bias-corrected nonparametric percentile bootstrap method [47]. The mediating and moderating effects are determined to be significant by testing whether the 95% confidence interval contains zero. If it does not contain zero, the effect is significant.

## 3. Results

### 3.1. Common Method Bias Test

All items in the questionnaire were analyzed by Harman’s one-factor analysis [48]. Our results revealed that a total of 13 factors had eigenvalues greater than 1. Of these, the first common factor had a variance contribution of 21.03%, which was below the critical value of 40%. Thus, the common method bias effect of this study was not significant.

### 3.2. Descriptive Statistics and Correlation Analysis of Each Variable

Results are tabulated in the Table 1 below. The MPA levels of college students were positively correlated with sleep quality, anxiety and ES (*p* < 0.001). The MPA of college students was negatively correlated with CR (*p* = 0.001). Anxiety was positively correlated with ES and sleep quality (*p* < 0.001), and was negatively correlated with CR (*p* < 0.001). CR was negatively correlated with sleep quality (*p* < 0.001), but was positively correlated with ES (*p* < 0.001). ES was positively correlated with sleep quality (*p* < 0.001). In addition, age was positively correlated with MPA and sleep quality (*p* < 0.001). In order to reduce the effect of spurious and undecomposed effects, gender and age were included as control variables in this study for subsequent analysis.

### 3.3. The Mediating Effect of Anxiety

We conducted a regression analysis of the data using gender and age as control variables to examine the mediating effect of anxiety between MPA and sleep quality. As shown in Table 2 below, MPA was significantly associated with sleep quality (*β* = 0.06, *p* < 0.001) and anxiety (*β* = 0.23, *p* < 0.001). When these variables were put into the regression model to predict sleep quality, MPA (*β* = 0.03, *p* < 0.001) and anxiety (*β* = 0.11, *p* < 0.001) were significantly positively correlated with sleep quality. This indicated that anxiety partially mediated the association between MPA and sleep quality (95% Bootstrap confidence interval [0.021, 0.030], and the mediating effect accounted for 41.67% of the total effect). See Table 2 and Table 3 for specific results.

### 3.4. Moderating Effect Test

First, we examined the moderating effect of CR, as shown in Table 4. MPA positively predicted anxiety (*β* = 0.22, *p* < 0.001), CR negatively predicted anxiety (*β* = −0.55, *p* < 0.001), and the interaction between MPA and CR also negatively predicted anxiety (*β* = −0.01, *p* = 0.002, *ΔR*^2^ = 0.01, *p* = 0.002). These suggested that the effects of MPA on anxiety were moderated by CR. To further investigate this moderating effect, the present study used the Johnson–Neyman method to quantify the role of CR in the relationship between MPA and anxiety [49]. See Figure 2A below for details. This indicates that CR negatively moderated the mediating effect of anxiety on the association between MPA and sleep quality. This means that with the increase in the CR level, the indirect effect of MPA on sleep quality through anxiety is weakened.

Secondly, this study examined the moderate role of ES. According to Table 4, MPA (*β* = 0.22, *p* < 0.001) and ES (*β* = 0.23, *p* < 0.001) positively predicted anxiety, and the interaction between MPA and ES was able to positively predict anxiety (*β* = 0.01, *p* = 0.001, *ΔR*^2^ = 0.01, *p* = 0.001). These results demonstrated that the effects of MPA on anxiety were moderated by ES. We also used the Johnson–Neyman method to quantitatively analyze the role of ES in the relationship between MPA and anxiety. The results are shown in Figure 2B. This indicates that ES positively moderated the mediating effect of anxiety on the relationship between MPA and sleep quality. It suggested that the indirect effect of MPA on anxiety was enhanced with the increase in the suppression level.

## 4. Discussion

Previous studies on MPA and sleep quality have concentrated more on the role that a single emotion plays. Based on the previous empirical studies, this research added emotion regulation as a variable. As expected, anxiety mediated the association between MPA and college students’ sleep quality. CR and ES played different moderating roles between MPA and anxiety. These findings help us to understand the mechanisms of sleep problems and provide a basis for preventing and improving students’ physical and mental health problems.

### 4.1. The Mediating Role of Anxiety

Our results show that anxiety significantly and positively predicts college students’ sleep quality. Furthermore, they further confirmed that anxiety partially mediates the relationship between MPA and sleep quality, which supports hypothesis 1. Previous studies have found that mobile phone addicts need to spend a lot of time and energy on mobile phone use every day, which can easily lead to negative emotions, such as anxiety [50]. In addition, symptoms such as headaches, wrist and neck pain, caused by the excessive use of smartphones, are also one of the reasons why people feel anxious [51,52,53]. According to the sleep disturbance process theory, excessive emotional or physiological arousal will seriously affect individuals’ sleep quality [22,54,55]. That is, mobile phone addicts are more likely to experience anxiety due to psychological or physiological reasons [16], and excessive anxiety will damage their sleep quality [21,56]. Therefore, anxiety is the mediating factor between MPA and sleep quality. This suggests that in order to improve college students’ sleep quality in the future, it is necessary not only to control the intensity of their smartphone use, but also to pay attention to their emotional states.

### 4.2. The Moderating Role of Cognitive Reappraisal

Consistent with hypothesis 2a of this study, the effect of MPA on anxiety was moderated by CR. There was a negative correlation between CR and anxiety, which was consistent with Goldin et al. [36]. Specifically, people with low CR use enhanced the predictive effect of MPA on anxiety. In contrast, those with high rates of CR use exerted a mitigating effect. In a sense, people who use CR strategies can reappraise negative emotional situations and events [57]. Thus, they reduce their negative emotional experiences and external expressions, which ultimately results in lower anxiety levels [58]. On the other hand, people have an instinct to seek benefits and avoid harm. According to resource conservation theory [34], individuals will adopt strategies to reduce their resource and energy consumption. Thus, mobile phone addicts with high CR use adopt positive emotion regulation strategies to offset or diminish the depletion of internal psychological resources and protect their resources [18,31]. In contrast, individuals with a low frequency of CR use tend to adopt negative emotion regulation strategies, which exacerbate anxiety [59]. This demonstrates that CR strategies can help patients control the generation and spread of negative emotions and alleviate individuals’ negative emotional experiences by protecting their internal resources. Finally, it can reduce anxiety levels and improve sleep quality [10]. Therefore, we should not only guide college students to use mobile phones rationally, but also pay attention to developing positive CR strategies.

### 4.3. The Moderating Role of Expressive Suppression

Furthermore, we also found that ES modulates the effects of MPA on anxiety, supporting hypothesis 2b. Specifically, ES was able to positively moderate the effect of MPA on anxiety. This suggests that ES may be a catalyst for the effects of MPA on anxiety. The effect of MPA on anxiety was more significant in individuals who used the ES strategy more frequently than in those who used it less frequently [39]. The reason for this may be that individuals who use ES strategies more frequently are not able to get rid of their negative emotions efficiently [32]. They are simply forced to suppress them. This results in stronger physiological responses and anxious emotional experiences [28]. In the end, it would lead to a worsened sleep quality for individuals [41]. It follows that the suppression and control of negative emotions cannot improve the emotional state of individuals, but that it is counterproductive. This suggests that, to some extent, ES will not only not release negative emotions, but may also increase bad emotions and damage the individual’s physical and psychological health [60]. As such, college students should appropriately vent their negative emotions to avoid more serious consequences.

In addition, the results of our study reveal for the first time that CR and ES have different moderating roles between MPA and anxiety. CR could mitigate the negative effects of MPA on anxiety, and have a more positive and active moderating effect [33]. However, there was no direct effect on sleep quality. ES was significantly correlated with individual MPA as well as anxiety levels [61]. This suggests that people with high MPA levels are more likely to adopt ES strategies and will also have higher levels of anxiety. ES would enhance the effects of MPA on anxiety. Furthermore, in agreement with the findings of Wang et al. [62], the results also demonstrated a positive correlation between ES and CR. It is clear from the results of this study that there are both differences and coexistences between CR and ES, and that the rational use of both can truly reflect the ability of emotion regulation [63]. Hence, when preventing and intervening in MPA and the sleep quality of college students, attention should be paid to both developing students’ positive emotion regulation skills and guiding them to minimize the use of ES strategies. In this way, the role of emotion regulation can be maximized.

### 4.4. Implications and Limitations

Most previous studies have examined the association between MPA and sleep quality from a single emotional perspective [10,16]. In this study, the role of emotion regulation in this relationship is explored from a higher perspective, at a top-down level. Our results reveal the effects of MPA on sleep quality in college students and the mechanisms that underlie the effects of anxiety and emotion regulation. On the one hand, at the individual level, college students should strengthen their emotional management and regulation ability. They should increase their positive emotional experiences and participate in more outdoor sports to maintain a positive and stable emotional state [64]. This would reduce the physical and mental problems caused by negative factors and weaken the negative effects of MPA on sleep quality [65]. On the other side, in terms of the living environment, parents and teachers should guide students to use cell phones in a reasonable and healthy way. They should take certain measures to control students’ internet time to prevent them from using cell phones excessively, and pay more attention to students’ psychological and physical health.

Furthermore, there are some limitations to the current study. On the one hand, the questionnaires used in the present study were self-report scales. In future, data support from third parties such as teachers, peers, or parents could be collected to improve the objectivity and ecological validity of the study. In addition, we noted that the “time window” was inconsistent across the original scales, and future studies could explore whether the consistency of the scale “time window” had an effect on the study results. On the other hand, in future studies, more demographic information, such as BMI, residence location and other factors that may have an impact on the study, should be included. Only in this way can we reveal the influence mechanism of the research more comprehensively. Finally, our research was a cross-sectional study, which only verifies the correlation between variables. Subsequent studies can be conducted longitudinally to explore the causality between variables.

## 5. Conclusions

This study found that MPA positively predicted sleep quality among college students and that anxiety mediated between them. In addition, CR plays a negative moderating role between MPA and anxiety. In contrast, ES plays a positive moderating role between MPA and anxiety.

## Figures and Tables

**Figure 1 behavsci-13-00250-f001:**
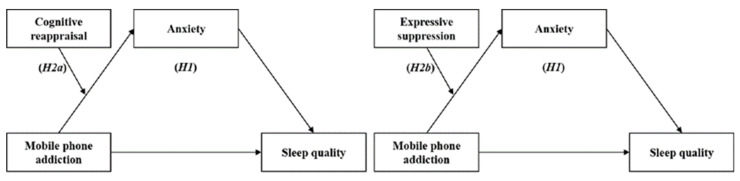
The hypothesized model of each variable.

**Figure 2 behavsci-13-00250-f002:**
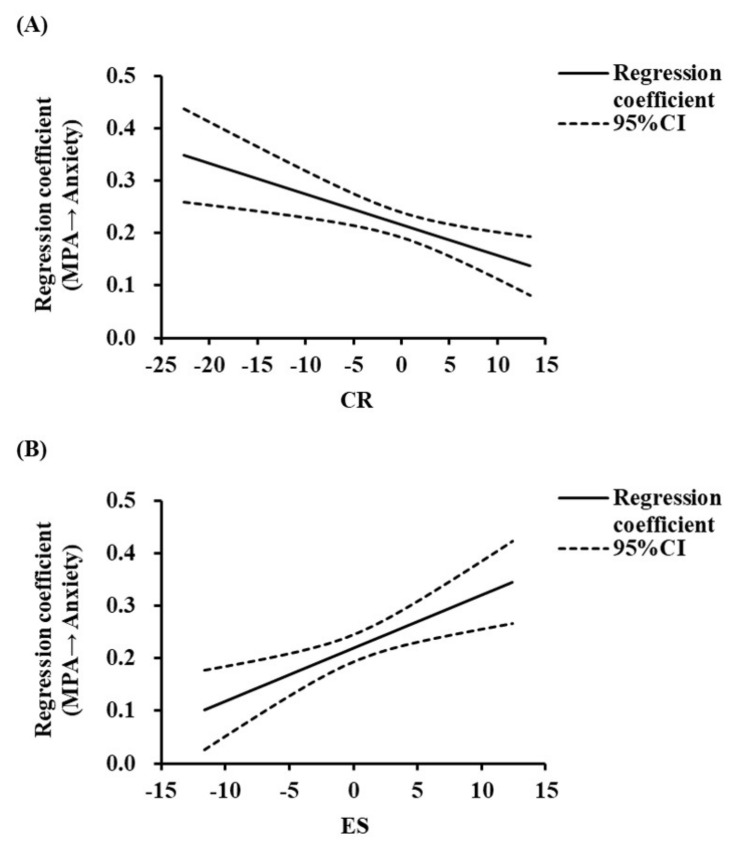
(**A**,**B**). The moderating effect of CR and ES on the relationship between MPA and anxiety. (**A**) shows the interaction between MPA and CR, and (**B**) shows the interaction between MPA and ES.

**Table 1 behavsci-13-00250-t001:** Means, standard deviations and correlation analysis among study variables (*N* = 1559).

Variables	M	SD	1	2	3	4	5	6
1. Gender	0.59	0.49	1					
2. Age	19.11	1.22	−0.10 ***	1				
3. MPA	53.41	17.03	0.01	0.10 ***	1			
4. Anxiety	41.55	9.62	0.02	0.04	0.41 ***	1		
5. CR	28.63	6.18	−0.01	−0.01	−0.08 **	−0.37 ***	1	
6. ES	15.61	3.97	−0.01	0.01	0.24 ***	0.18 ***	0.09 ***	1
7. Sleep quality	8.04	2.76	0.02	0.06 *	0.37 ***	0.46 ***	−0.13 ***	0.16 ***

Note: Gender: 0 = male, 1 = female; * *p* < 0.05, ** *p* < 0.01, *** *p* < 0.001.

**Table 2 behavsci-13-00250-t002:** Regression analysis of the mediating effect of anxiety in the relationship between MPA and sleep quality.

Dependent Variables	Independent Variables	R^2^	F	*β*	t
Sleep quality	Gender	0.14	83.20 ***	0.12	0.87
	Age			0.06	1.19
	MPA			0.06	15.54 ***
Anxiety	Gender	0.17	105.75 ***	0.25	0.56
	Age			−0.03	−0.16
	MPA			0.23	17.73 ***
Sleep quality	Gender	0.25	131.58 ***	0.09	0.72
	Age			0.07	1.34
	MPA			0.03	8.89 ***
	Anxiety			0.11	15.45 ***

Note: *** *p* < 0.001.

**Table 3 behavsci-13-00250-t003:** Test for mediating effects.

Effect	Value	95% CI	
		Lower	Upper
Indirect effect (MPA → Anxiety → Sleep quality)	0.025	0.021	0.030
Direct effect (MPA → Sleep quality)	0.035	0.027	0.043
Total effect	0.060	0.052	0.067

**Table 4 behavsci-13-00250-t004:** Regression analysis of the moderating effect of CR and ES on the mediating effect of anxiety.

Dependent Variables	Independent Variables	R^2^	F	*β*	t
Sleep quality	Gender	0.25	131.58 ***	0.09	0.72
	Age			0.07	1.34
	MPA			0.03	8.88 ***
	Anxiety			0.11	15.45 ***
Anxiety	Gender	0.29	125.16 ***	0.13	0.30
	Age			−0.05	−0.31
	MPA			0.22	17.69 ***
	CR			−0.55	−16.02 ***
	MPA × CR			−0.01	−3.03 **
Sleep quality	Gender	0.25	131.58 ***	0.09	0.72
	Age			0.07	1.34
	MPA			0.03	8.88 ***
	Anxiety			0.11	15.45 ***
Anxiety	Gender	0.18	69.32 ***	0.32	0.71
	Age			−0.03	−0.16
	MPA			0.22	16.35 ***
	ES			0.23	4.01 ***
	MPA × ES			0.01	3.27 **

Note: ** *p* < 0.01, *** *p* < 0.001.

## Data Availability

The data that support the findings of this study are available in the [figshare] repository (https://doi.org/10.6084/m9.figshare.21652283.v1), accessed on 1 December 2022.

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
