# Peer review of "Mobile Phone Addiction and Sleep Quality: The Mediating Role of Anxiety and the Moderating Role of Emotion Regulation"

_behavsci, 2023, doi:10.3390/bs13030250_

Round 1

Reviewer 1 Report (Previous Reviewer 2)

The authors use a cross-sectional study to study if anxiety might mediate the effect of mobile phone addiction (MPA) on sleep quality. With the data available to the authors it is not possible to address this, since it is possible that sleep quality might have affected mobile phone addiction or the mediator anxiety. This is a major limitation of the study that limits the possibility to study the role of MPA on sleep quality as well as to determine the mediating role of anxiety in this association

Author Response

Dear Reviewer,

Thank you very much for your valuable comments and careful review of our manuscript. We have responded to your comments point by point. Please see the attachment.

Sincerely,

Yours

Reviewer 2 Report (New Reviewer)

The main concern of this article is related to the concept of mediation used and its conceptual, methodological and statistical justification. The authors should further justify these three aspects.

It is easy to understand and assume that personal characteristics on how people self-regulate (cognitive reappraisal and suppression) are constituted as variables of individual differences in sleep quality. It is more doubtful to consider anxiety as a mediating variable since in clinical studies it is usually considered as an outcome variable. When it is stated in the discussion that mobile phone addiction generates negative emotions such as anxiety, an interdependence between predictor and mediator variables is being pointed out, which may distort the concept of mediation. Even more so when the interrelation between the mediating variable (anxiety) and the dependent variable (sleep quality) is clear.  The evidence is incontrovertible that poorer sleep quality generates anxiety and greater anxiety generates poorer sleep quality regardless of whether or not one is addicted to cell phones. I believe that these issues should be clarified in relation to the concept of mediation and the questionable use of anxiety as a mediating variable.

https://www.annualreviews.org/doi/abs/10.1146/annurev.psych.58.110405.085542

https://www.frontiersin.org/articles/10.3389/fpsyg.2017.01984/full

https://pubmed.ncbi.nlm.nih.gov/17716046/

Author Response

Dear Reviewer,

Thank you very much for your valuable comments and careful review of our manuscript. We have responded to your comments point by point. Please see the attachment.

Sincerely,

Yours

Reviewer 3 Report (New Reviewer)

Dear Authors,

Thank you for the opportunity to review an interesting article entitled: ‘Mobile Phone Addiction and Sleep Quality: The Mediating Role of Anxiety and the Moderating Role of Emotion Regulation’. The aim of the research presented in this article is to analyse the mediating role of anxiety and the moderating role of emotion regulation on the relationship between mobile phone addiction and sleep quality among students. The topics are extremely relevant due to the use of smartphones by increasingly younger people.

The strengths of the article presented for evaluation are the solid theoretical background, the large sample size and the use of statistical analysis.

The reviewer's job, on the other hand, is to help improve the article so that it meets the highest possible standards of the journal, therefore I will focus on its weaknesses.

Introduction

[1].  The lack of a clearly formulated research objective, which should come before the formulated hypotheses.

Method

[2].  Please provide the Cronbach's α coefficients of the scales used in your study, not just the Cronbach's α coefficient for your study.

[3].  Please provide the authors of the original 'Emotion Regulation' questionnaire, not just the authors of the adaptation.

Results

[4].  In line 191, shouldn't it be "Table 2" instead of "Figure 2"?

[5].  Table 4 is above, not below, the description from lines 216-224, hence the "below" from line 216 is best removed - there is no need to indicate the position of the tables relative to their description, since the authors use table/line numbers in the text.

References

[6].  DOI numbers should be completed in the bibliography.

Author Response

Dear Reviewer,

Thank you very much for your valuable comments and careful review of our manuscript. We have responded to your comments point by point. Please see the attachment.

Sincerely,

Yours

This manuscript is a resubmission of an earlier submission. The following is a list of the peer review reports and author responses from that submission.

Round 1

Reviewer 1 Report

In their study, the authors focus on the important and complex topic of the consequences of overusing mobile devices. Interestingly, the work doesn't just take into account the interdependence of phone addiction and sleep quality and anxiety but also consider top-down emotion regulation. This research proposes to construct anxiety as a mediating variable and emotion regulation as a moderating variable.  

This study found negative moderating role of cognitive Reappraisal between MPA and anxiety and positive moderating role of expressive Suppression between MPA and anxiety. Understanding these mechanisms in the context of phone addiction may allow the development of optimal strategies for dealing with the effects of phone addiction. Further studies are needed however to explore the causality between these variables.

The study is well designed and clearly presented. The results are supported by good statistical analysis. The results and conclusions are presented in a clear and structured way. The authors also described the limitations of the study.

Author Response

Dear Reviewer,

Thanks very much for taking your time to review this manuscript. We really appreciate all your comments and suggestions. Attached to this letter is our point-by-point response to the comments raised by you. These comments are copied, while our responses are given directly after in red color. Please see the attachment.

Thanks again!

Reviewer 2 Report

The authors used data from a cross-sectional study to study the association between mobile phone addiction and sleep quality and also aim to evaluate if anxiety mediated this association. However, the main limitation of this study is that the authors are using data from a cross-sectional study and it is therefore not possible to study mediation. This is because it is not possible to know if mobile phone addiction has lead to anxiety (anxiety is a potential mediator) or anxiety has affected mobile phone use/addiction (anxiety is a potential confounder) since data is available also at one point. Assessing whether cognitive reappraisal (CR) and expressive repression (ES) moderate the association is also not possible since data is only available for one time point.

My additional comments to the authors are the following:

in the introduction the authors write "screen light and emitted electromagnetic radiation from electronic devices affect the secretion of melatonin in individuals, which interferes with sleep rhythms and affects sleep quality" and are citing a study by Green et al (reference 12) that however is only on screen light exposure. Please revise the text or provide additional references.

The authors should provide more information regarding the "time window" that was used for defining mobile phone addiction. Did the questions for assessing mobile phone addiction refer to the last week, last month, last three months, or something else? The same information should also be reported for the other variables (anxiety, emotion regulation, and sleep quality).

How was MPA defined in this study? Is this scale specific for Chinese students? Has it been validated for other group of students/individuals?

The authors should provide more details on the statistical methods used in this paper and inform the readers about the statistical method that was used. Writing "This study examined the mediating effect using Model 4 of the PROCESS macro, and examined the moderating effect of CR and ES using model 7" is helpful only for SPSS users.

Age was included in the regression models but why was sex not included? There could be also other variables, for example BMI, that could be linked to both mobile phone use and sleep.

What was the range of the MPA, sleep quality, and anxiety scales among the study participants? Did the authors use sleep quality as a binary outcome or as a continuous outcome? This is not clear from the methods section: the authors have mentioned some cut-off value for the sleep quality but then it is not clear if that was used or not. If sleep quality was used as a continuous variable, then a one-unit change on the MPA scale is associated with a 0.06 increase on the sleep quality scale. The authors should comment on whether this is a large or a small effect and also on whether this is clinically relevant. Results would be more easier to understand if mobile phone addiction, anxiety, and sleep quality would be dichotomized.

The authors should provide more descriptive characteristics of the study population. Moreover, the titles used for the tables are not very informative.

There a few typos in the text and tables, for example MPA in table 4  is mispelled as MPS

Author Response

Dear Reviewer,

Thanks very much for taking your time to review this manuscript. We really appreciate all your comments and suggestions. Your suggestions have enabled us to improve our work. Attached to this letter is our point-by-point response to the comments you have made. These comments are copied, while our responses are given directly after in red color. Please see the attachment.

Thanks again!

Reviewer 3 Report

I read with great interest the Manuscript titled “Mobile Phone addiction and Sleep Quality: The mediating Role of Anxiety and the moderating Role of emotion Regulation” In my opinion, the topic is interesting enough to attract the readers’ attention.

Authors should consider the following recommendations

Material and Methods: The authors should describe Inclusion and exclusion criteria.

Did the authors used social media for the online recruitment?

Line 32, Line 71, Line 116, Line 121, Line 185: please correct the minor English errors  

Author Response

Dear Reviewer,

Thanks very much for taking your time to review this manuscript. We really appreciate all your comments and suggestions. Your suggestions have led to improvements in our work. Attached to this letter is our point-by-point response to the comments you have made. These comments are reproduced, while our responses are given directly after in red. Please see the attachment.

Thanks again!

Reviewer 4 Report

This is a large-scale cross-sectional study focusing on mobile phone addiction and sleep quality, and the mediating role of anxiety and the moderating role of emotion regulation. I have some concerns about the article.

1、They included participants of 17 years old, who were juveniles in China, so were the informed consent got from their parents? 

2、I'm interested in the subgroup analysis of gender and singleton or non-singleton.

Author Response

Dear Reviewer,

We would like to thank you for the time and effort you have put into reviewing the previous version of the manuscript. Your suggestions have enabled us to improve our work. Attached to this letter is our point-by-point response to the comments you have made. These comments are copied, while our responses are given directly after in red color. Please see the attachment.

Thanks again!

Reviewer 5 Report

The authors describe  correlation analysis of self-reported measures of mobile phone addiction and sleep quality, and looking specifically at the mediating effect of anxiety and moderation by emotional regulation.

This is a very specific hypothesis that is difficult to assess in isolation. The cross-sectional design, self-reported information and absence of the potential to adjust for confounders prohibits drawing meaningful conclusions. This is an issue of study design and the authors wont be able to address these through a revision of the manuscript.

Author Response

(The authors gave the same response as above.)

Round 2

Reviewer 2 Report

With the additional information provided by the authors, I have now understood that the mediator "anxiety" is assessed in the week preceding the questionnaire, while the questions regarding "sleep quality" (the outcome of the study) refer to the past month. It is therefore possible that a low sleep quality in the past month might affect anxiety in the week before the questionnaire was filled in. The mediator should be measured temporally before the outcome: it is therefore not possible to study mediation with this type of data.

From the statistical methods it is not clear which statistical method was used, was it a linear regression model? The authors have only mentioned the SPSS command that was used to evaluate the association between MPA and sleep quality

I do not agree with the authors´ decision of adjusting only for age. I believe that sex is an important confounder as girls tend to have a higher risk of anxiety than boys, especially at that age. The list of confounders that should be included in the regression model should be decided a priori rather than because of the association observed in the data.

The authors are still not mentioning in the manuscript that all variables (including the outcome and the exposure) were used as continuous. This is an important detail that should be mentioned in the methods section.

Reviewer 4 Report

The article was well-written and the comments were addressed properly.

Reviewer 5 Report

Unfortunately, the problems identified in the first version of the manuscript are the result of the study design, not the manuscript. This cannot simply be solved by adding it as a limitation.